# Comparative Efficacy of Various Exercise Therapies and Combined Treatments on Inflammatory Biomarkers and Morphological Measures of Skeletal Muscle among Older Adults with Knee Osteoarthritis: A Network Meta-Analysis

**DOI:** 10.3390/biomedicines12071524

**Published:** 2024-07-09

**Authors:** Che-Li Lin, Hung-Chou Chen, Mao-Hua Huang, Shih-Wei Huang, Chun-De Liao

**Affiliations:** 1Department of Orthopedic Surgery, Shuang Ho Hospital, Taipei Medical University, New Taipei City 235041, Taiwan; 2Department of Orthopedics, School of Medicine, College of Medicine, Taipei Medical University, Taipei 11031, Taiwan; 3Department of Physical Medicine and Rehabilitation, Shuang Ho Hospital, Taipei Medical University, New Taipei City 235041, Taiwan; 4Department of Physical Medicine and Rehabilitation, School of Medicine, College of Medicine, Taipei Medical University, Taipei 11031, Taiwan; 5Department of Biochemistry, University of Washington, Seattle, WA 98015, USA; 6Department of Physical Medicine and Rehabilitation, Wan Fang Hospital, Taipei Medical University, Taipei 116, Taiwan; 7International Ph.D. Program in Gerontology and Long-Term Care, College of Nursing, Taipei Medical University, Taipei 110301, Taiwan

**Keywords:** osteoarthritis, sarcopenia, exercise, inflammation, muscle hypertrophy

## Abstract

Osteoarthritis is associated with high risks of sarcopenia in older populations. Exercise interventions are promising treatments for musculoskeletal impairments in knee osteoarthritis (KOA). The purpose of this study was to identify the comparative effects of exercise monotherapy and its adjunct treatments on muscle volume and serum inflammation for older individuals with KOA. A literature search in the electronic databases was comprehensively performed from this study’s inception until April 2024 to identify relevant randomized controlled trials (RCTs) that reported muscle morphology and inflammation outcomes after exercise. The included RCTs were analyzed through a frequentist network meta-analysis (NMA). The standard mean difference (SMD) with a 95% confidence interval was estimated for treatment effects on muscle morphology and inflammation biomarkers. The relative effects on each main outcome among all treatment arms were compared using surface under the cumulative ranking (SUCRA) scores. The certainty of evidence (CoE) was assessed by the GRADE (Grading of Recommendations, Assessment, Development, and Evaluations) ranking system. Probable moderators of the treatment efficacy were investigated by network meta-regression analysis. This study included 52 RCTs (4255 patients) for NMA. Among the 27 identified treatment arms, isokinetic training plus physical modality as well as low-load resistance training plus blood-flow restriction yielded the most optimal treatment for inflammation reduction (−1.89; SUCRA = 0.97; CoE = high) and muscle hypertrophy (SMD = 1.28; SUCRA = 0.94; CoE = high). The patient’s age (β = −0.73), the intervention time (β = −0.45), and the follow-up duration (β = −0.47) were identified as significant determinants of treatment efficacy on muscle hypertrophy. Exercise therapy in combination with noninvasive agents exert additional effects on inflammation reduction and muscle hypertrophy compared to its corresponding monotherapies for the KOA population. However, such treatment efficacy is likely moderated by the patient’s age, the intervention time, and the follow-up duration.

## 1. Introduction

Knee osteoarthritis (KOA) is one of the most prevalent joint disorders among older individuals, and it may develop from degenerated articular structures and deficits in the skeletal muscle system [1]. KOA is a whole joint disease accompanied by pathological signs in all joint tissues, including the menisci, subchondral bone, infrapatellar fat pad, cartilage, and synovial membrane. Since osteoarthritis has been recognized as an inflammatory disease, the pathogenesis of KOA is not only derived from locally inflamed joint synovial membranes and infrapatellar fat pads but its disease severity can be further mediated by systemic inflammaging [2], a condition that refers to the low-grade inflammation accompanied by physiological aging. Such low-grade persistent chronic inflammation also plays a critical role in driving the development and disease progression of sarcopenia in older individuals [3,4,5], especially those who have chronic diseases such as diabetes [6,7,8], arthritis [8,9,10,11], and cardiopulmonary diseases [8,12]. Under such a scenario, the intrinsic molecular mechanisms of KOA may crosstalk with sarcopenia, in terms of an increased catabolic reaction evoked by proinflammatory mediators during muscle protein turnover [5], despite the unclear causal relationship between these two diseases [13].

Muscle weakness in older adults with KOA is a prominent risk factor contributing to physical limitations while the disease progresses [14,15]. Declining muscular strength in older individuals has been attributed to decreased muscle mass as well as morphological changes, which can be termed sarcopenia and atrophy, respectively [1,16]. Under such a threat of aging-related muscle attenuation, older individuals with KOA face substantial risks of sarcopenia and muscle atrophy [17,18,19]. Low muscle mass has been associated with the radiographic disease stage of KOA [20,21,22] as well as with severe symptoms [23] and the impairment of functions [24]. Therefore, the development of effective interventions to promote muscle hypertrophy in older individuals with KOA who have high risks of sarcopenia is vital to prevent muscle weakness and physical limitations.

Exercise therapy has been believed to exert promising benefits with respect to muscle mass [25] and strength [26] gains, pain reduction [27,28,29,30,31], and function recovery [26,27,28,29,30,31] for older people with KOA. Because exercise allows older individuals to work against the increased muscle anabolic resistance that occurs with advancing age [32,33], rationally, it is beneficial for restoring muscle volume in the older population, especially those who have high sarcopenia risks [34,35]. Additionally, exercise exerts clinical effects on muscle hypertrophy by activating muscle protein synthesis and promoting the proliferation of muscle satellite cells [36], on which skeletal muscle size enlargement and myofiber regeneration are highly dependent [37,38]. However, the relative effects of different exercise types on muscle hypertrophy in KOA remain unclear. Although a number of researches have reported that patients with KOA showed increased changes in muscle morphology after exercise [39,40,41], some studies have concluded that exercise exerted no evident effects to trigger structural remodeling in skeletal muscle, which results in muscle hypertrophy [42,43,44]. Given the facts that both sarcopenia and KOA share the common feature of increased serum inflammation, which suppresses skeletal muscle growth and homeostasis [45,46], and that training-induced changes in serum inflammation and muscle volume contribute to strength gains after exercise [47,48], determining the effects of exercise on inflammation reductions and muscle size increases can help clinical practitioners develop appropriate intervention strategies for older individuals with KOA, especially those who have high risks of sarcopenia.

Despite the treatment efficacy of various exercise therapies for KOA investigated by the previous systematic review studies [25,26,27,28,29,30,31,49,50,51,52,53,54,55,56,57,58], most of them have targeted the treatment outcomes of pain relief, strength gain, and function recovery [26,28,29,30,31,49,50,51,52,53,54,55,56,57,58], and few have examined the training effects on muscle hypertrophy [25] and inflammation reduction. In addition, one systematic review has compared different exercise types with respect to their effects on serum inflammatory biomarkers among older people who were healthy or had chronic conditions [59]; however, the exercise types were limited to resistance exercise training (RET), aerobic exercise training (AET), and a combination of both, and the older individuals with KOA were not included. Taken together, the relative effects on inflammation and muscle hypertrophy among other different exercise types such as isokinetic exercise training (IKET) remains unclear. Therefore, the objectives of this network meta-analysis (NMA) study were to identify (1) the comparative effects among monotherapy and combined regimens of exercise on changes in serum inflammation level and muscle volume for older individuals with KOA; (2) the most superior exercise type for promising treatment efficacy; and (3) any substantial moderators for training efficacy.

## 2. Materials and Methods

### 2.1. Study Protocol and Design

This NMA study adapted the Preferred Reporting Items for Systematic Reviews and Meta-Analysis (PRISMA) Extension Statement for NMA [60]. The protocol of this systematic review was registered in the PROSPERO registry (CRD42024545904).

### 2.2. Search Strategy

We comprehensively searched electronic online databases—namely PubMed, the Physiotherapy Evidence Database (PEDro), EMBASE, the Cochrane Library, the China Knowledge Resource Integrated Database, CINAHL, and Google Scholar—from this study’s inception to April 2024. The manual searches were additionally performed in relevant systematic reviews for eligible references. There was no limitation applied to the publication language. We used an internet-based collaboration platform, the Covidence electronic workflow platform [61], which diminishes the study selection process in a systematic review study [62], to screen and assess the identified studies.

The criteria for Participant, Interventions, Comparisons, Outcomes, and Study Design (PICOS) were presented in Table 1. The following search terms were used for participants’ conditions: (“middle-aged adults” OR “older adults” OR “elder adults”) AND “osteoarthritis”. The following keywords were used for the intervention: “exercise training” OR “physical activity”. The following keywords were used for the main outcome: “inflammatory markers/pro-inflammatory cytokine” OR “cross-sectional area/muscle volume/muscle thickness”. Appendix A shows the detailed search terms in all databases.

### 2.3. PICOS Criteria for Study Selection

Trials were included if they met the following criteria (Table 1): (1) the trial design was a randomized control trial (RCT) with an experimental group (i.e., exercise) and a control group, which was published in a peer-reviewed or scientific journal; (2) participants were identified as having KOA with a radiographic or symptomatic diagnosis; (3) the exercise was employed as the primary treatment method or as adjunctive therapy in the experimental group; (4) the comparator group received another exercise or relatively lower intensity of exercise training compared with the experimental group. The control group receiving regular care (RC), not related to exercise, served as a reference in this study; and (5) the trial reported the main outcome measures of inflammatory biomarkers and muscle volume (described below). Two authors (CDL and SWH) individually and independently performed procedures, including the study search, screening, and selection of relevant articles. Where any disagreement occurred, other team members (CLL) were consulted for judgment until a consensus was reached.

### 2.4. Outcome Measures

The main outcomes of interest included the systemic inflammation level and muscle volume. Systemic inflammation was assessed by serum biochemical analysis and was expressed as inflammatory indicators including C-reactive protein and proinflammatory cytokines such as interleukins and tumor necrosis factors. Muscle volume was measured by an imaging methodology such as magnetic resonance imaging, and ultrasound and was expressed as morphological measures including muscle cross-sectional area, muscle thickness, and thigh volume.

### 2.5. Data Extraction

An evidence table was established to extract the following data from each included study (Table 1): (1) characteristics of the study arm, according to the intervention nature; (2) characteristics of the participants, including age, body mass index, sex distribution, and disease severity, using the Kellgren–Lawrence osteoarthritis grading system (KL grade) as well as the Outerbridge osteoarthritis grade; (3) characteristics of the exercise training protocol; (4) measured time points; and (5) main outcome measures. According to the training intensity, the RET was classified as high with high load (HIRET, ≥80% one repetition maximum), moderate load (MIRET, 30–80% one repetition maximum), and low load (LIRET, ≤30% one repetition maximum). The cutoff points were selected based on the American College of Sports Medicine’s (ACSM) recommendations for exercise intention in older individuals [63]. The follow-up time interval over the study period was assessed and defined as short (<6 months), medium (≥6 months or <12 months), and long (≥12 months); where multiple follow-up time points were reported, the longest time period was selected for analysis.

Initially, one reviewer (CDL) extracted relevant data from the included studies, followed by another reviewer (SWH) who reviewed the extracted data. Where any disagreement occurred, a consensus meeting was conducted to resolve it, and a third reviewer (CLL) was further consulted if the disagreement persisted.

### 2.6. Risks of Bias and Methodological Quality in Individual Study and across Studies

The Cochrane risk of bias tool [64,65], incorporating the PEDro rating scale, which assesses the methodological quality of the analyzed studies [66], was employed to evaluate the bias risk and methodological quality of the analyzed studies [67].

Assessments and judgments of the risk of bias for the included studies were separately performed by two researchers (CDL and SWH). If any difference of opinion existed, it was resolved during a consensus meeting. A third reviewer (HCC) was invited, if necessary, to judge the persistent inconsistency.

The following seven bias domains, in correspondence with 13 rating items, which were derived from the PEDro scale [66] and the rated biases in the estimates of intervention effects [68], were assessed and judged: (1) selection bias, including rating items of random sequence generation, allocation concealment, and similarity at baseline; (2) performance bias, associated with rating items of blinding of participants and personnel, blinding of therapists or care providers; (3) detection bias, associated with blinding of the outcome assessor; (4) attrition bias, which corresponded with incomplete outcome data; (5) reporting bias, raised from selective reporting; (6) unconscious bias, raised from disclosures of the authors’ conflicts of interest, and other sources of potential bias such as funding or sponsorship bias (i.e., sponsor or funding resource and its role). The conflicts of interest as well as the role of the sponsor are likely to have an impact on study results, and this raises research-agenda bias, while counting for overall results [69]. Each of the 13 rating items was ranked as a low, high, or unclear risk with respect to the indicated bias, in accordance with the criteria of judgment. Counting the methodological quality overall, each included trial was classified as having a low, moderate, or high risk of bias [65].

We assessed the publication bias by visual inspection of funnel plots and by performing the Begg–Mazumdar rank correlation test [70].

### 2.7. Data Synthesis and Analysis

In light of the variations in measurement tools across trials, the treatment effects in each of the included RCTs were estimated and expressed as standard mean differences (SMDs), along with 95% confidence intervals (CIs). The mean change score and the corresponding standard deviation (SD) of each outcome measure were directly extracted from each study arm of the included RCTs. Where the SD of the change score of the outcome measure was not available, the SDs of the baseline (*SD_pretest_*) and post-intervention (*SD_posttest_*) were used to estimate the missing SD of the change score by the following calculation:$${SD}_{change}=\surd [({SD}_{pretest}{)}^{2}+({SD}_{posttest}{)}^{2}\;\;\;\;\;\;\;\;\;\;\;\;\;\;\;\;\;-2\times {Corr}_{pretest–posttest}\;\times \;{SD}_{pretest}\;\times \;{SD}_{posttest}]$$

Based on the recommendation of the Cochrane Handbook for Systematic Reviews of Interventions [71], the correlation coefficient between the baseline score and post-intervention score (*Corr_pretest–posttest_*) can be available from the included studies, which have reported the SD change for the outcome measure. Following Rosenthal’s method, we used a *Corr_pretest–posttest_* value of 0.7 for a conservative estimation [72].

A pairwise comparison and frequentist random-effects NMA modeling were performed to estimate direct and indirect effects across all treatment arms for each main outcome. By employing the I^2^ statistic alongside τ^2^ values, heterogeneity, as well as global consistency, were assessed to estimate variance across the comparisons. Consistency between direct and indirect comparisons was assessed by the node-splitting approach, and the results were expressed by a forest plot [73]. Additionally, the total inconsistency across all treatment arms was assessed using a decomposition method, which establishes a full design-by-treatment interaction model [74]. The ranking of multiple competing treatments per outcome was performed and expressed as the surface under the cumulative ranking (SUCRA) score [75].

In order to explore any relevant moderators that explain the heterogeneity across comparisons, we established network meta-regression (NMR) models. Possible moderators were identified as follows: (1) participant characteristics, including age, female sex proportion, body mass index, disease onset duration, world region of study application, and disease severity (i.e., percentage number of patients with KL grade ≥ 3 in the study) and (2) the study methodology, including treatment composition (i.e., monotherapy or combined treatment), risk of bias, training duration, and follow-up duration. The NMR results were reported as *β* along with a 95% credible interval (CrI).

The treatment compliance in response to exercise interventions was measured by the all-cause withdrawal rate. In addition, any adverse events reported by the included RCTs were extracted, and the adverse effects were evaluated by calculating the occurrence rate of adverse events. The analysis results were presented as odds ratios (ORs) and 95% CIs.

All NMA and NMR analyses were performed by the R statistical software (version 4.4.0, R Foundation for Statistical Computing, Vienna, Austria) [76,77]. The statistical significance was assumed at *p* < 0.05 for all the analyses.

### 2.8. Certainty of Evidence

The quality of evidence contributing to NMA estimates were assessed by the Grading of Recommendation Assessment, Development, and Evaluation (GRADE) approach for each main outcome [78]. The GRADE framework has been employed to evaluate the strength of evidence in systemic reviews [79]. It evaluates five categories of each treatment arm in the included RCTs, including study limitations, inconsistency, imprecision, incoherence, and publication bias. Based on the ranked score of each category, which was considered to upgrade or downgrade the quality of evidence, the certainty of evidence was judged as high, moderate, low, and very low overall. Two reviewers (CDL, SWH) independently performed the ranking and grading procedures, which were followed by the others (HCC, MHH, CLL), to validate the initial grading results.

## 3. Results

### 3.1. Selection of Studies

The PRISMA procedure of the study selection process is presented in Figure 1. Initially, we identified a total of 871 potentially eligible articles after electronic searches and manual screening of the literature. After excluding 430 duplicate articles, we screened the retrieved titles and abstracts of 441 articles to assess their eligibility, followed by the full-text review of 102 relevant articles. Finally, 52 RCTs [39,40,80,81,82,83,84,85,86,87,88,89,90,91,92,93,94,95,96,97,98,99,100,101,102,103,104,105,106,107,108,109,110,111,112,113,114,115,116,117,118,119,120,121,122,123,124,125,126,127,128,129] published between 1995 and 2024 were included in this NMA.

### 3.2. Characteristics of Analyzed Patients

Table 2 summarizes the demographic data and study characteristics of the included RCTs. A total of 4255 patients with KOA were recruited. Overall, the sample had a mean age of 63.3 (range, 53.2–76.7) years and a mean body mass index of 30.2 (range, 22.5–38.0) kg/m^2^.

Among 52 included RCTs, 41 enrolled both women and men with the average proportion of female participants being 63.7% (range, 19.7–93.3%), whereas 11 had sex-specific study designs (9 included only women [39,84,91,94,98,107,115,116,121]; 2 included only men [40,104]) and 2 RCTs [90,127] did not report the sex distribution of the study sample.

All of the included RCTs had a radiographic diagnosis of KOA and assessed the disease severity by the KL grade, but one [84] used the Outerbridge osteoarthritis grade. The majority of included RCTs had patients with mild to moderate KOA (i.e., KL grade 1–3), whereas 6 RCTs [94,100,114,124,125,129] recruited patients with mild KOA (i.e., KL grade ≤ 2) only, and 13 RCTs [81,85,89,92,93,96,108,109,112,117,118,122,123] enrolled patients with severe KOA (i.e., KL grade = 4).

### 3.3. Exercise Intervention Protocol

A summary of exercise protocols is presented in Table 2. Ten types of exercise training were identified (Table 2), namely, AET, RET, aquatic exercise therapy (AQET), IKET, isometric exercise training (IMET) performed by self-activated muscle contraction and by neuromuscular electrical stimulation (NMES), which passively activates muscle contraction, mind–body therapy (MBT), multicomponent exercise training (MET), comprising strength and function training, proprioceptive training (PropT), and whole-body electromyostimulation (WB-EMS). WB-EMS refers to an application of electrical stimulations for main muscle groups during body movement tasks [130].

The combined treatments included blood-flow restriction, electromyographic and ultrasound biofeedback, nutrition and diet interventions, physical agent modality, traditional Chinese medicine, and whole-body vibration.

In total, 27 treatment arms were identified for NMA. Each of the identified treatment regimens and its abbreviation is listed in Table 3.

### 3.4. Risks of Bias in Individual Studies and across STUDIES

The rank and detail on each risk-of-bias item, judged by the reviewing authors in each included RCT, are presented in Appendix A, respectively. An overall summary of the included RCTs is presented in Figure 2. The results showed that the majority of the included RCTs were judged as having an overall moderate or high risk of bias, whereas only five RCTs [86,88,89,90,120] were ranked as possessing high methodological quality along with a low risk of biases. In general, selection, performance, and attrition biases had major contributions in counting risks of biases across the included RCTs.

#### 3.4.1. Selection Bias

Insufficient information on random sequence generation and allocation concealment led to selection bias in the included RCTs (Appendix A). A total of 28 out of 52 RCTs (53.8%) [85,87,88,92,93,95,96,98,99,100,101,102,104,106,107,109,111,112,116,117,118,120,121,122,123,124,126,129] reported the randomization procedure, and 15 out of 52 RCTs (28.8%) [85,87,88,92,95,98,104,106,107,112,116,117,118,121,123] employed concealed allocation (Figure 2).

#### 3.4.2. Masks for Performance and Detection Biases

It is difficult to mask participants and therapists for group allocations when exercise was administered as a primary intervention or a cointervention without placebo controls, which were deemed as the major sources of performance bias among the included RCTs (Appendix A). Only 8 (15.4%) [39,98,102,113,114,116,118,121] and 4 (7.7%) [95,98,118,120] out of 52 RCTs successfully blinded the participants and therapists for the treatment assignment, respectively (Figure 2).

In order to minimize detection biases in assessing treatment outcomes, the assessor was blinded in approximately 32 out of 52 (62.5%) included RCTs [39,40,80,81,84,85,86,87,88,89,92,94,95,97,98,104,105,107,108,109,112,113,114,116,117,118,120,121,122,123,127,128] (Appendix A). However, 4 RCTs [96,99,101,106] that did not conduct a blinding procedure for the assessors were judged as having a high risk of detection bias, and the other 16 (30.8%) had unclear risks of detection bias. Accordingly, the overall summary showed a moderate risk of detection bias across RCTs.

#### 3.4.3. Attrition Bias

Inadequate follow-up patients and analysis without an intention-to-treatment method raised great contributions in the risk of attrition bias within numerous RCTs (Appendix A) and across studies (Figure 2). A total of 6 out of 52 RCTs (11.5%) [39,80,81,92,94,122] had a drop-out rate > 15%, and 19 (36.5%) [80,81,87,93,94,98,104,105,107,108,112,114,116,117,118,121,122,123,128] did not perform an intention-to-treatment analysis (Appendix A). Therefore, the overall risk of attrition bias was counted as moderate across the included studies despite all RCTs performing between-group comparisons.

#### 3.4.4. Research-Agenda Bias

Information on funding sources and the authors’ conflicts of interest disclosures is summarized in Appendix A. There were remarkably unclear risks of unconscious and funding bias due to no disclosures of competing interest (22 RCTs, 42.3%) and the unclear role of sponsors or funding resources (36 RCTs, 69.2%) within most of the included RCTs. Two RCTs [102,113] that obtained grants from the funding sources without a definite role and one [110] supported by the sponsor being involved in the intervention design were considered as having high risks of agenda bias. Overall, an unclear risk of agenda bias was counted across the included studies (Figure 2).

#### 3.4.5. Publication Bias

A visual inspection of the funnel plots of muscle hypertrophy and inflammation outcomes did not reveal a substantial asymmetry (Figure 3). Begg and Mazumdar’s rank correlation test indicated no evident publication bias among the trials reporting measures of muscle volume (*p* = 0.20) and inflammatory biomarkers (*p* = 0.51).

### 3.5. Effectiveness of Treatment for Muscle Hypertrophy

The direct and indirect comparative effects of all treatment arms (i.e., 20 exercise regimens and RC), compared with each other, are presented in Appendix A. The direct comparison results revealed that MIRET, IMET (Active), IMET (NMES), IMET (NMES) + PAM, and LIRET + BFR obtained greater changes in muscle volume, with standardized mean differences (SMDs) of 0.62 [95% CI: 0.01, 1.23], 1.15 (95% CI: 0.02, 2.27), 0.67 [95% CI: 0.07, 1.27], 1.15 (95% CI: 0.33, 1.97), and 1.72 (95% CI: 0.87, 2.58), respectively, compared to RC. In addition, the combined treatments LIRET + BFR exhibited superior effects on morphological increases in muscle compared with LIRET (SMD = 0.98; 95% CI: 0.19, 1.78).

Figure 4 presents the network connections among treatment arms for the treatment outcome of muscle hypertrophy. The NMA was based on 26 RCTs, which reported muscle morphological measures, corresponding with 21 treatment options and 49 pairwise comparisons. Considering RC as a reference, the combined options—namely, LIRET + BFR, IMET (NMES) + PAM, IMET (NMES), MIRET, and HIRET—obtained favorable effects on muscle hypertrophy, with significant SMDs of 0.62–1.28 (all *p* < 0.05), regardless of intervention and follow-up time (Figure 5). The global heterogeneity was significant across treatment arms (τ^2^ = 0.19, *I*^2^ = 61.2%, *p* = 0.0004). The Q statistic results assessing total consistency between designs in the NMA were insignificant under the assumption of a full design-by-treatment random-effects model (τ^2^ = 0.08, *p* = 0.06). The node-splitting analysis results showed no inconsistencies between the direct and indirect estimates (Appendix A).

Among all treatment arms, the composite LIRET + BFR was ranked the most effective (SUCRA = 0.94) for muscle hypertrophy—followed by IMET (NMES) + PAM (SUCRA = 0.79) and then IMET (Active) + BioF (SUCRA = 0.77; Figure 5).

### 3.6. Effectiveness of Treatment for Serum Inflammation

Direct and indirect comparative effects of all treatment arms (i.e., 19 exercise regimens and RC), compared with each other, are presented in Appendix A. The direct comparison results revealed that combined treatments IKET + PAM (SMD = −1.35), IMET (Active) + TCM (SMD = −1.61), IMET (Active) + WBV (SMD = −1.14), and MET + TCM (SMD = −1.19) obtained greater reductions in serum levels of inflammation compared to RC. In addition, the combined treatments IKET + PAM (SMD = −1.98) and IKET + TCM (SMD = −1.14) exhibited superior effects on decreasing inflammation compared with IKET.

Figure 6 presents the network connections among treatment arms for the treatment outcome of inflammation reduction. The NMA was based on 30 RCTs, which reported measures of inflammatory biomarkers, corresponding with 20 treatment options and 48 pairwise comparisons. Considering RC as a reference, the combined options—namely, IKET + PAM, IKET + TCM, IMET (Active) + TCM, MET + TCM, and AET—obtained favorable effects on inflammation reduction, with significant SMDs of −0.84 to −1.89 (all *p* < 0.05) during an entire follow-up timeframe (Figure 7). The global heterogeneity was significant across treatment arms (τ^2^ = 0.20, *I*^2^ = 76.7%, *p* < 0.0001). The full design-by-treatment interaction random-effects model showed significant inconsistency across all treatment arms (τ^2^ = 0.09, *p* = 0.04). In addition, the node-splitting results for NMA consistency revealed no inconsistencies between the direct and indirect evidence; similar findings were observed through a visual inspection of a forest plot (Appendix A).

Among all treatment arms, the composite LIRET + BFR was ranked the most effective (SUCRA = 0.94) for inflammation reduction—followed by IMET (NMES) + PAM (SUCRA = 0.79) and then IMET (Active) + BioF (SUCRA = 0.77; Figure 5).

Among all treatment arms, the composite IKET + PAM was ranked the most effective (SUCRA = 0.97) for inflammation reduction—followed by IKET + TCM (SUCRA = 0.90) and then IMET (Active) + TCM (SUCRA = 0.84; Figure 7).

### 3.7. Network Meta-Regression Analyses Results

The NMR results are presented in Table 4. The analysis results revealed that the patient’s age (β = −0.73), the follow-up duration (β = −0.47), and the treatment time duration (β = −0.45) were significantly associated with the SMDs for muscle hypertrophy. According to such results, a young age tends to obtain great short-term treatment effects during a short period of intervention. There was no significant moderator for treatment efficacy on inflammation reduction.

### 3.8. Certainty of the Evidence

Table 5 summarizes the GRADE certainty ratings for each main outcome. Overall, the certainty of the evidence was judged as low, moderate, and high among the combined treatments for treatment effects on muscle volume and inflammation. The composite LIRET + BFR achieved a high certainty of evidence in increasing muscle volume. In addition, TCM as well as PAM in combination with exercises obtained a high certainty of evidence in decreasing serum levels of inflammation, particularly MET, IMET (Active), and IKET. The most common judgments downgrading the certainty were associated with major concerns about study limitations, imprecision (i.e., a wide range of 95% CI), and the small sample sizes of the studies.

### 3.9. Compliance and Adverse Effects

All-cause dropout and adverse effects are summarized in Appendix A. As a whole, the 52 included RCTs reported an attrition rate ranging from 0% to 53.8% during the study period, of which 7.1% to 25.0% were eliminated in five RCTs [39,81,85,92,121] because of exercise-related noncompliance, particularly the HIRET [39,81], MIRET [92,121], MIRET + PAM [121], LIRET + BFR [92], IMET (NMES) [81], and MET + ND [85] (Appendix A). Treatment compliance was calculated based on the all-cause withdrawal. There was no difference in compliance across all the treatment arms while RC served as the reference (Figure 8A).

No serious adverse event occurred after exercise training (Appendix A). However, three RCTs [39,92,111] reported mild to moderate knee pain after resistance and isokinetic training. No significant all-cause adverse effects were observed among all of the treatment regimens while RC served as the reference (Figure 8B).

## 4. Discussion

### 4.1. Summary of Main Findings

The primary objective of this study was to compare the relative efficacy among different treatment regimens of exercise training for serum inflammation and muscle volume outcomes in older adults with KOA. The main results showed that (1) by direct comparisons, a combined treatment of exercise training and a noninvasive agent yields additional effects compared to exercise alone, particularly the composites LIRET + BFR, IKET + PAM, and IKET + TCM; and (2) using the SUCRA method in NMA, the composites LIRET + BFR and IKET + PAM were ranked as the most effective strategy, with a high certainty of evidence for increasing muscle volume and reducing serum inflammation, respectively, among the identified treatment arms. Further, the NMR results revealed that (1) a patient’s age may affect the relative treatment efficacy among treatment arms and (2) follow-up time, and the intervention period may have influences on treatment efficacy, particularly on the muscle hypertrophy outcome.

### 4.2. Comparisons of this NMA with Previous Studies

In order to identify optimal types of exercise for older individuals with KOA, several systematic reviews have compared relative efficacy among different exercise regimens [25,27,31,49,50,51,52,53,54,55,56,57,58,131,132,133,134]. Most of the previous studies focused on pain reduction [27,31,49,51,53,54,55,56,57,58,132,133], strength gain [55,57,131,133,134], and function restoration [27,31,49,50,52,54,55,56,57,58,132,133]. However, treatment effects on muscle mass or volume increases have been less discussed [25,131]. In addition, most of the previous studies generally classified the exercise regimens into four types, namely, strengthening [25,26,27,28,29,31,50,51,52,55,56], flexibility [31,56,58], aerobic [27,31,51,52,56], or aquatic [31,51,52,53,58], which may not entirely categorize other specific forms of exercise, such as IMET [26,56], IKET [26,135,136,137], BFR training [131,132,133], and NMES [49,134]. Furthermore, mind–body exercises such as yoga and tai chi have yet to be comprehensively compared with conventional exercise therapies [27,50,51,54]. In the present study, the treatment outcome with respect to muscle size increases was targeted as the primary outcome, and the eligible RCTs were included accordingly. A total of 10 exercise types (i.e., monotherapy) and 15 combined regiments were identified from the included RCTs (Table 3), and all of the treatment arms were completely compared in an NMA (Figure 4 and Figure 6). In line with previous conclusions [31], the analytic results in our current study showed that combined regimens of exercise training and adjunct treatments generally achieved superior treatment effects compared with monotherapies. We further identified that the combined treatment LIRET + BFR is most likely the optimal treatment option for muscle hypertrophy as well as IKET + PAM for inflammation reduction, in KOA.

A previous systemic review investigated the treatment efficacy of exercise for KOA, and the results demonstrated that isotonic training exerts significant effects on increasing muscle volume (SMD = 0.88; *p* < 0.00001), whereas IMET and IKET subgroups do not [25]. Consistent with the previous results, the NMA results in this study indicated that HIRET (i.e., isotonic training) obtained favorable effects on muscle hypertrophy (SMD = 0.71; 95% CI: 0.08–1.34), whereas IMET and IKET did not with reference to RC (Figure 5). We further identified that HIRET (SUCRA = 0.67) and MIRET (SUCRA = 0.47) are superior in the rank of optimal treatments compared with IMET (SUCRA = 0.46) or IKET (SUCRA = 0.32; Figure 5).

### 4.3. Moderator of Relative Efficiency among Treatment Regimens

Treatment effects in response to exercise regimens are likely affected by the patient’s age. Our findings indicated that a younger age predicts greater treatment effects of exercise therapies on muscle hypertrophy. Results in the present study can be supported by the previous researchers who concluded that younger patients (<60 years) may benefit more from exercise therapy compared with those aged ≥ 60 years, particularly in terms of the pain outcome.

### 4.4. Strengths and Limitations

This study fully compared the relative effects among multiple exercise monotherapies as well as their adjunct regimens, particularly the identified 10 types of exercise, in older individuals with KOA.

This NMA has some limitations. First, the PAM comprised a variety of treatment agents and physical modalities, which were employed by the included RCTs. Therefore, it was difficult to identify a definite PAM protocol for treatment efficacy in the main outcome. Secondly, a total of 6 out of 52 analyzed RCTs [81,90,91,94,108,122] enrolled a small study sample (n < 30); thus, these studies that reported insignificant treatment effects on main outcomes may contribute a negative effect size to the overall pooled result.

Finally, most of the included RCTs investigated outcomes within a 6-month follow-up. Only four RCTs [80,85,109,112] conducted a long-term follow-up duration (i.e., 12 months or longer). It remains unclear whether treatment effects were maintained for a time period of 12 months or longer. Further studies are warranted to understand the long-term effects of exercise in older patients with KOA, especially the muscle morphology and inflammatory outcomes.

## 5. Conclusions

The present NMA determined the comparative efficacy among multiple exercise monotherapies and their combined treatments for older patients with KOA. The composite LIRET + BFR as well as IKET + PAM was determined to be the most effective treatment option for muscle hypertrophy and inflammation reduction, respectively, regardless of the intervention protocol or follow-up timeframe. The findings of the present study may help guide clinicians to prescribe a prompt exercise regimen to ensure optimal treatment outcomes for KOA in older individuals.

## Figures and Tables

**Figure 1 biomedicines-12-01524-f001:**
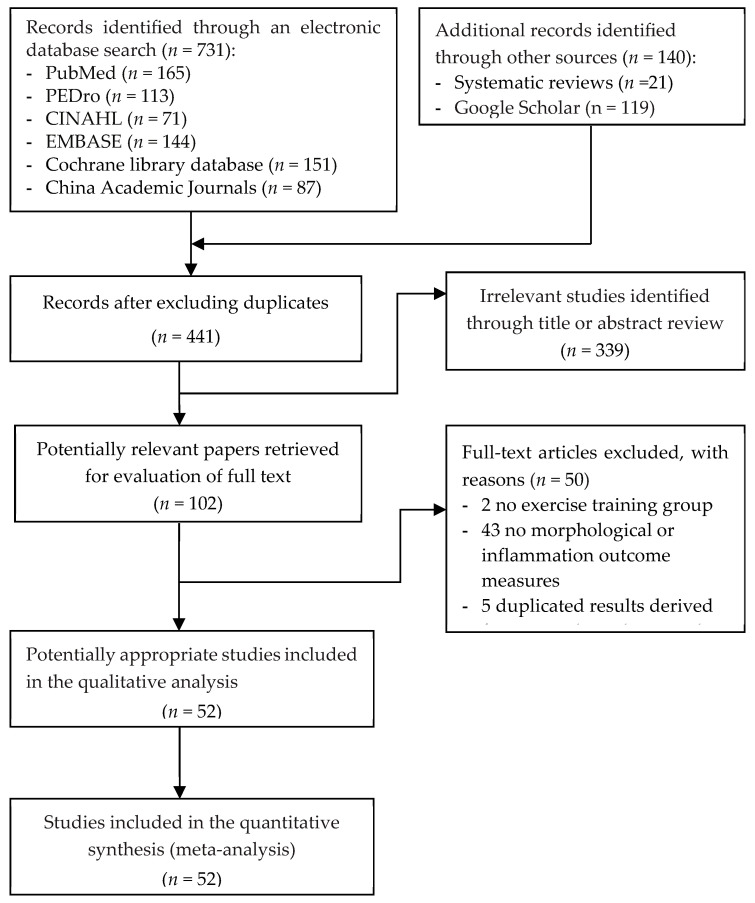
PRISMA flowchart of the study selection.

**Figure 2 biomedicines-12-01524-f002:**
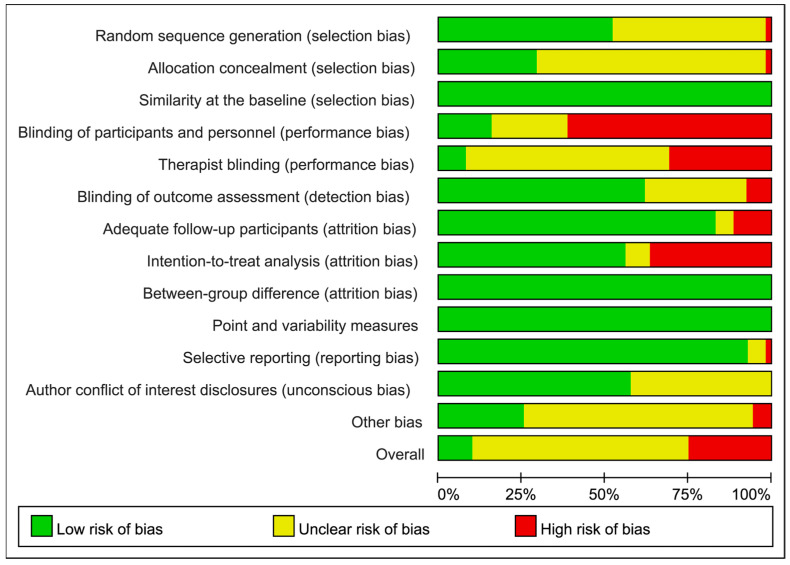
Risk of bias summary: review authors’ judgments about each risk of bias item presented as percentages across all included studies.

**Figure 3 biomedicines-12-01524-f003:**
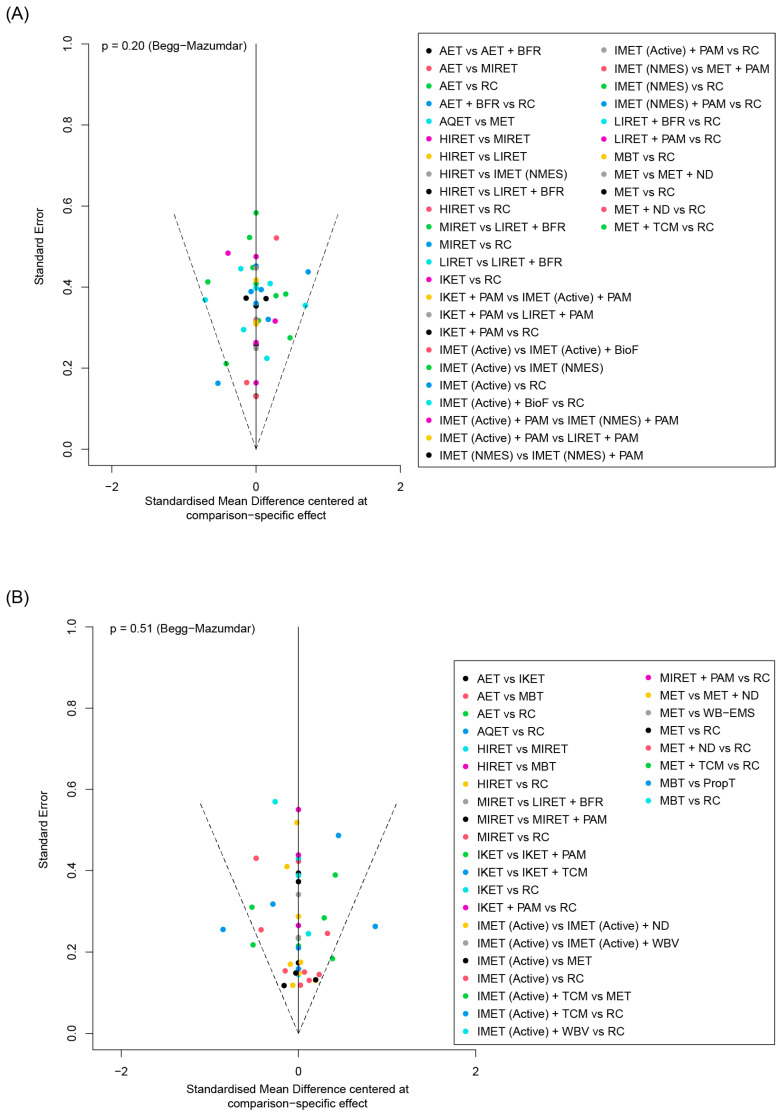
Funnel plot for identifying publication bias in (**A**) muscle hypertrophy and (**B**) inflammation outcomes. AET, aerobic exercise training; AQET, aquatic exercise therapy; BFR, blood flow restriction; BioF, biofeedback; HIRET, high-intensity resistance exercise training; IKET, isokinetic exercise training; IMET, isometric exercise training; LIRET, low-intensity resistance exercise training; MBT, mind–body therapy; MET, multicomponent exercise training; MIRET, medium-intensity resistance exercise training; ND, nutrition and diet; NMES, neuromuscular electrical stimulation; PAM, physical agent modality; PropT, proprioceptive training; RC, regular care; TCM, traditional Chinese medicine; WB-EMS, whole-body electromyostimulation; WBV, whole-body vibration.

**Figure 4 biomedicines-12-01524-f004:**
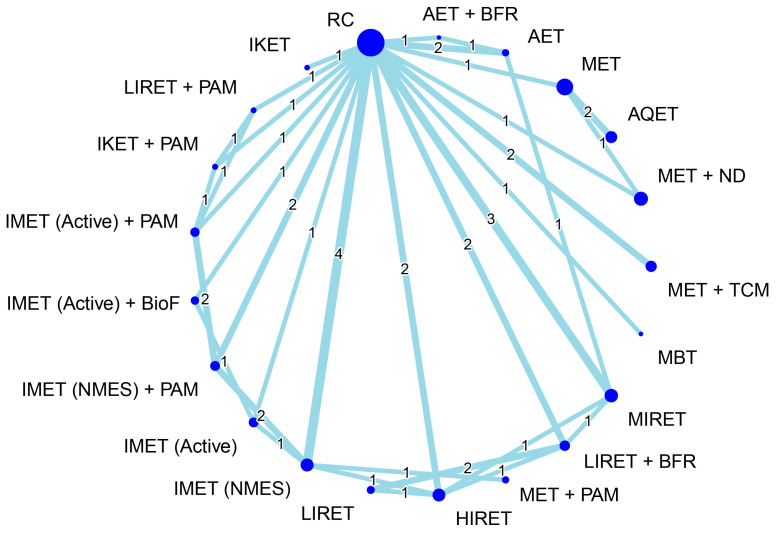
Geometric network diagram demonstrating the interconnections among different treatment arms for muscle volume. Each node represents a treatment arm, and the size of the node is proportional to the number of participants assigned to the specific treatment. The thickness of each line is proportional to the number of direct comparisons between arms denoted on the line. AET, aerobic exercise training; AQET, aquatic exercise therapy; BFR, blood flow restriction; BioF, biofeedback; HIRET, high-intensity resistance exercise training; IKET, isokinetic exercise training; IMET, isometric exercise training; LIRET, low-intensity resistance exercise training; MBT, mind–body therapy; MET, multicomponent exercise training; MIRET, medium-intensity resistance exercise training; ND, nutrition and diet; NMES, neuromuscular electrical stimulation; PAM, physical agent modality; RC, regular care; TCM, traditional Chinese medicine.

**Figure 5 biomedicines-12-01524-f005:**
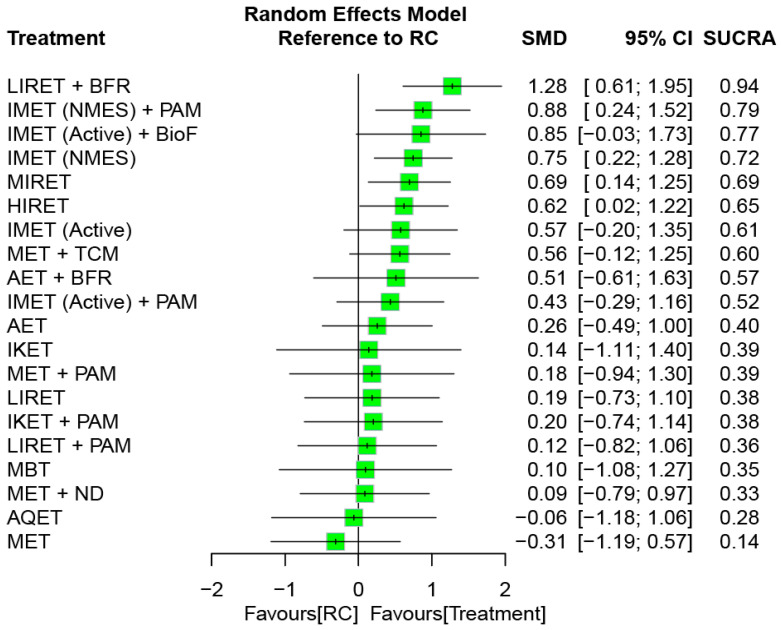
Relative efficacy of treatment regimens on muscle hypertrophy for an entire follow-up duration. SMD, standardized mean difference; CI, confidence interval; SUCRA, surface under the cumulative ranking curve; AET, aerobic exercise training; AQET, aquatic exercise therapy; BFR, blood flow restriction; BioF, biofeedback; HIRET, high-intensity resistance exercise training; IKET, isokinetic exercise training; IMET, isometric exercise training; LIRET, low-intensity resistance exercise training; MBT, mind–body therapy; MET, multicomponent exercise training; MIRET, medium-intensity resistance exercise training; ND, nutrition and diet; NMES, neuromuscular electrical stimulation; PAM, physical agent modality; RC, regular care; TCM, traditional Chinese medicine.

**Figure 6 biomedicines-12-01524-f006:**
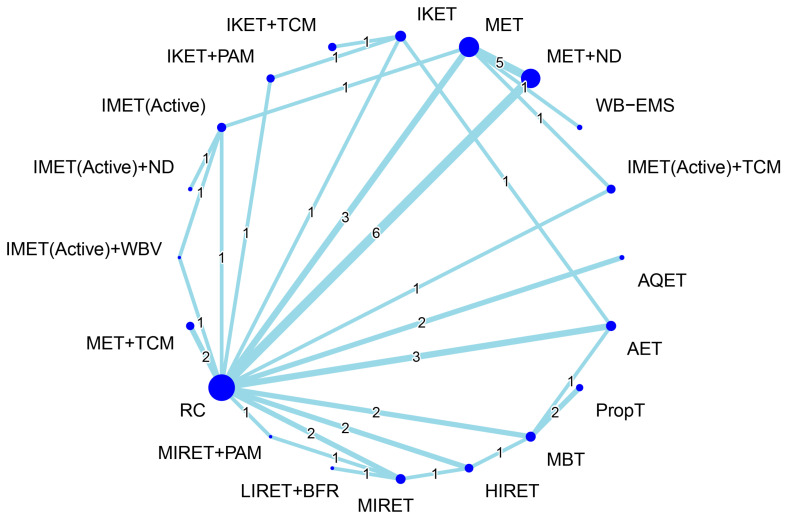
Geometric network diagram demonstrating the interconnections among different treatment arms for inflammation reduction. Each node represents a treatment arm, and the size of the node is proportional to the number of participants assigned to the specific treatment. The thickness of each line is proportional to the number of direct comparisons between arms denoted on the line. AET, aerobic exercise training; AQET, aquatic exercise therapy; BFR, blood flow restriction; HIRET, high-intensity resistance exercise training; IKET, isokinetic exercise training; IMET, isometric exercise training; LIRET, low-intensity resistance exercise training; MBT, mind–body therapy; MET, multicomponent exercise training; MIRET, medium-intensity resistance exercise training; ND, nutrition and diet; PAM, physical agent modality; PropT, proprioceptive training; RC, regular care; TCM, traditional Chinese medicine; WB-EMS, whole-body electromyostimulation; WBV, whole-body vibration.

**Figure 7 biomedicines-12-01524-f007:**
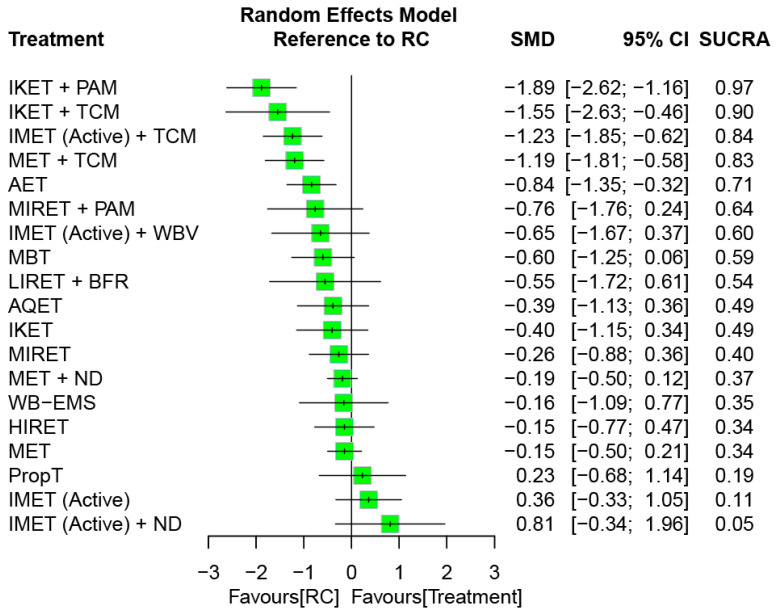
Relative efficacy of treatment regimens on inflammation reduction for an entire follow-up duration. SMD, standardized mean difference; CI, confidence interval; SUCRA, surface under the cumulative ranking curve; AET, aerobic exercise training; AQET, aquatic exercise therapy; BFR, blood flow restriction; HIRET, high-intensity resistance exercise training; IKET, isokinetic exercise training; IMET, isometric exercise training; LIRET, low-intensity resistance exercise training; MBT, mind–body therapy; MET, multicomponent exercise training; MIRET, medium-intensity resistance exercise training; ND, nutrition and diet; PAM, physical agent modality; PropT, proprioceptive training; RC, regular care; TCM, traditional Chinese medicine; WB-EMS, whole-body electromyostimulation; WBV, whole-body vibration.

**Figure 8 biomedicines-12-01524-f008:**
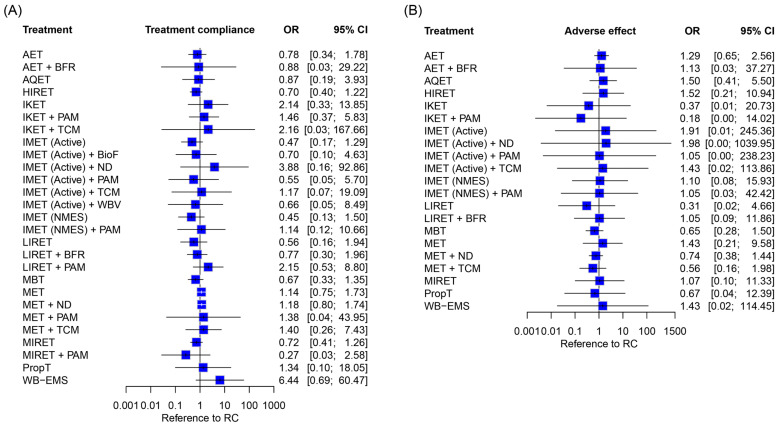
Participant compliance and side effects of exercise regimens. Data concerning all-cause (**A**) withdrawals and (**B**) adverse events were pooled using inverse variance weighting methods. OR, odds ratio; CI, confidence interval; AET, aerobic exercise training; AQET, aquatic exercise therapy; BFR, blood flow restriction; BioF, biofeedback; HIRET, high-intensity resistance exercise training; IKET, isokinetic exercise training; IMET, isometric exercise training; LIRET, low-intensity resistance exercise training; MBT, mind–body therapy; MET, multicomponent exercise training; MIRET, medium-intensity resistance exercise training; ND, nutrition and diet; NMES, neuromuscular electrical stimulation; PAM, physical agent modality; PropT, proprioceptive training; RC, regular care; TCM, traditional Chinese medicine; WB-EMS, whole-body electromyostimulation; WBV, whole-body vibration.

**Table 1 biomedicines-12-01524-t001:** PICOS criteria for study selection.

Participant (P)
1. Age: 50 years or older
2. Having a diagnosis of knee osteoarthritis
Intervention (I)
1. Monotherapy: any exercise therapy alone. The specified exercise therapies are defined as follows:
(1) Resistance-based exercise training: Any exercise that requires muscles to contract against an external resistance force throughout an entire joint movement. The extra force can be free weights (e.g., dumbbells or barbells), elastic bands or tubing, self-body weight, or any other object that causes counter actions of the muscles.
(2) Isometric exercise training: Any exercise that requires the muscles to contract against an external resistance force without joint movement. The muscle contraction can be actively initiated or passively activated by neuromuscular electrical stimulation.
(3) Isokinetic exercise training: An exercise that requires the muscles to contract at a constant speed of joint movements, irrespective of the amount of resistance applied. It is termed isokinetic contraction.
(4) Aerobic exercise training: an exercise that refers to cardiovascular conditioning such as walking, running, swimming, and bicycling.
(5) Aquatic or water-based exercise training: an exercise that refers to therapeutic motion regimens performed in a water environment
(6) Proprioceptive or sensory-motor training: An exercise aimed at improving balance and reducing fall risk in older adults. It is also termed kinesiotherapy, which challenges an individual’s ability of postural control to stabilize a joint during static or dynamic functional tasks.
(7) Mind–body therapy: An exercise approach that combines body movement, breathing control, and attention focus to improve physical and overall health such as yoga, tai chi, and qigong. It emphasizes the relationships among a person’s mental, physical, and spiritual experiences.
(8) Whole-body electromyostimulation: an application of electrical stimulations for main muscle groups during body movement tasks.
(9) Muti-component exercise regimen: an exercise regimen composed of two or more of the exercise types listed above.
2. Combined treatment: an exercise therapy in combination with an adjunct noninvasive treatment. The adjunct treatment included the following:
(1) Physical assistant agents (such as biofeedback, blood-flow restriction, whole-body vibration) or modalities (such as therapeutic ultrasound diathermy, interferential current, and transcutaneous electrical nerve stimulation)
(2) Traditional Chinese medicine (such as herbal medicine and electric acupoint therapy)
(3) Nutrition or diet intervention (such as protein supplementation, weight loss, or an energy-restricted program)
Comparison (C)
The comparator included the following:
1. Different exercise type
2. Lower training intensity
3. Regular care without any exercise training
Outcome (O)
1. Inflammation biomarker
(1) C-reactive protein
(2) Pro-inflammatory cytokine (a) Interleukins (IL), such as IL-1, IL-6, IL-8, and IL-1β (b) Tumor necrosis factor (TNF) family, such as TNF-α, TNF-α receptor, and tumor necrosis-like weak inducer of apoptosis
2. Measures of muscular morphology
Cross-sectional area, muscle volume, muscle thickness
Study design (S)
1. Randomized parallel (two-arm or multiple-arm) controlled trial
2. Randomized crossover trial

**Table 2 biomedicines-12-01524-t002:** Summary of included study characteristics.

Study Year	Country (Area)	Study Arm	Age (Years) ^a^	BMI (kg/m^2^) ^a^	Sex		N	K-L Grade	Disease Duration (Month)	Exercise Intervention			Outcome Measures ^b^						Follow- up Time (Week)
										Frequency (Session /Week)	Duration (Week)	Adherence (%)	Muscle Morphology			Inflammatory Biomarker			
					Female	Male							CSA	MT	TVol	Interleukin	TNF	CRP	
Beavers	USA	MET + ND	65.5	33.5	108	43	151	2, 3	NR	3	72	70			v	v		v	0, 24,
2015 [80]		MET	65.5	33.5	108	42	150					66							72
		RC	65.8	33.7	105	44	149					61							
Bruce-Brand	Ireland	HI-RET	63.4	33.9	4	6	10	3, 4	NR	3	6	83.3	v						0, 6
2012 [81]		IMET (NMES)	63.9	33.7	4	6	10					91.3							
		RC	65.2	31.7	3	3	6												
Chen	China	IMET (Active) + TCM	65.8	NR	27	19	46	1–3	58.3	7	4	NR				v	v		0, 4
2021 [82]		MET	66.3		26	20	46		60.4										
Chen	China	IMET (Active) + TCM	61.2	NR	23	27	50	1–3	33.7	7	5	NR				v	v		0, 5
2023 [83]		RC	60.9		35	25	60		33.7										
Choi	Korea	IMET (Active) + BioF	72.8 ^c^	24.3 ^c^	20	0	20	2.5 ^d^	NR	3	8	NR		v					0, 8
2015 [84]		RC			10	0	10												
Christensen	USA	MET + ND	67.4	36.5	52	12	64	≥1	114	4	52	13.8						v	0, 52
2013 [85]		RC	66.4	37.8	103	25	128		96.0			61.5							
Cook	USA	MI-RET	76.7	26.8	7	5	12	NR	NR	2	12	95	v						0, 6,
2017 [86]		LI-RET + BFR	76.5	26.8	7	5	12					96							12
		LI-RET	74.8	26.2	7	5	12					100							
de Almeida	Brazil	AET	55.6	26	15	5	20	2, 3	27.1	3	14	92 ^c^	v						0, 14
2020 [87]		MI-RET	55.2	26	16	5	21		26.6										
		RC	53.8	27	16	4	20		24.2										
Devrimsel	Turkey	IMET (NMES)	61.2	30	23	7	30	2, 3	69.1	5	3	100		v					0, 3
2019 [88]		MET + PAM	62.8	28.9	24	6	30		83.5			96.7							
Ferraz	Brazil	HI-RET	59.9	30.3	16	0	16	2, 3	52.8	2	12	90	v						0, 12
2018 [39]		LI-RET + BFR	60.3	30.2	16	0	16		52.8			91							
		LI-RET	60.7	29.9	16	0	16		56.4			85							
Franz	Germany	AET + BFR	61.5	26.8	3	7	10	3, 4	NR	2	6	100			v				0, 3,
2022 [89]		AET + PLA	64.2	27.7	3	7	10					100							6
		RC	66.3	29.4	4	6	10												
Gur	Turkey	IKET	55.5	31.3	NR		17	2, 3	NR	3	8	100	v						0, 8
2002 [90]		RC	57.0	32.3			6					100							
Ha	China	AQET	60.9	25.2	9	0	9	NR	NR	3	4	NR						v	0, 4
2018 [91]		RC	61.3	24.6	8	0	8												
Harper	USA	LI-RET + BFR	67.2	31.7	10	6	16	≥2	NR	3	12	81.4					v		0, 12
2019 [92]		MI-RET	69.1	29.8	15	4	19					83							
Ji	China	MET + TCM	65.2	NR	12	18	30	≥2	37.4	7	4					v	v		0, 4
2016 [93]		RC	64.3		10	20	30		38.9										
Jiang	China	MBT	64.2	25.8	11	0	11	1, 2	47.0	5	12	NR		v					0, 12
2020 [94]		RC	62.9	22.9	12	0	12		42.6										
Kelmendi	Germany	WB-EMS	58.3	31.1	22	14	36	2, 3	NR	1.5	29	88				v		v	0, 29
2024 [95]		MET	57.9	29.5	24	12	36					>90							
Kim	USA	AQET	67.4	32.9	10	10	20	3, 4	NR	3	12	96.6				v	v	v	0, 12
2021 [96]		RC	66.9	31.9	9	14	23												
Kocaman	Turkey	IMET (NMES) + PAM	61.6	29.4	15	4	19	1–3	NR	5	4	NR	v						0, 4
2008 [97]		IMET (Active) + PAM	60.4	31.2	14	5	19												
Kuntz	Canada	HI-RET	63.7	28.9	11	0	11	NR	NR	3	12	90				v	v	v	0, 12
2018 [98]		MBT	65.5	30.1	10	0	10					100							
		RC	71.1	32.3	10	0	10					90							
Li	China	MBT	63.7	23.9	22	8	30	2, 3	NR	3	12	NR				v	v		0, 12
2022 [99]		PropT	62.7	24.2	23	5	28												
Lin	China	AET	65	NR	45	52	97	1, 2	15.7	5	24	NR				v	v		0, 24
2022 [100]		RC	65.4		16	18	34		16										
Liu	China	AET	55	23.4	23	4	27	2, 3	9.0	5	12	93				v			0, 12
2019 [101]		MBT	54.5	23.0	46	11	57		6.5			93.5							
		RC	55	23.4	14	10	24		6.5			97							
Lu	China	MET + TCM	64.9	NR	30	34	64	2, 3	24.7	1~5	16	NR				v			0, 16
2022 [102]		RC	64.9		28	36	64		24.1										
Ma	China	IKET + TCM	56.3	NR	42	50	92	2, 3	8.7	2	8	NR				v	v	v	0, 8
2019 [103]		IKET	57.1		41	52	93		8.9										
Mahmoud	KSA	IMET (Active)	54.6	35.0	0	32	32	2, 3	42.0	3	12	NR		v					0, 12
2017 [40]		RC	53.2	34.8	0	12	12		45.6										
Mahmoud	KSA	LI-RET + BFR	60.2	30.8	0	17	17	2, 3	59.9	3	8	85	v						0, 8
2021 [104]		RC	59.1	29.7	0	18	18		60.1			90							
Malas	Turkey	IKET + PAM	56.2	31.8	51 ^c^	10 ^c^	20	1–3	49.2	5	5	NR		v					0, 5
2013 [105]		IMET (Active) + PAM	61.2	33.2			22		58.8										
		LI-RET + PAM	59.1	30.0			19		54.0										
		RC	58.9	33.2			61		54.2										
McLeod	USA	MET + ND	66.5	33.7	64	8	72	NR	NR	3	8	64				v	v	v	0, 8
2020 [106]		MET	67.2	33.9	73	10	83					70							
Melo Mde	Brazil	IMET (NMES) + PAM	69.6	29.0	14	0	14	2, 3	≥6	2	8	NR	v	v					0, 8
2015 [107]		IMET (NMES)	69.3	33.0	15	0	15												
		RC	67.7	30.0	15	0	15												
Messier	USA	MET + ND	67	35.0	10	3	13	≥2	NR	3	24	94.7				v			0, 24
2000 [108]		MET	69	38.0	7	4	11					82.6							
Messier	USA	HI-RET	67	31.0	52	75	127	≥2	NR	3	72	66			v				0, 72
2021 [109]		MI-RET	64	31.0	51	75	126					69							
		RC	64	32.0	48	76	124					80							
Miller	USA	MET + ND	69.8	34.9	20	11	31	NR	NR	3	24	77.5				v	v	v	0, 24
2008 [110]		RC	69.5	34.4	20	16	36												
Mu	China	IKET + PAM	55	22.5	23	32	55	2, 3	NR	5	4	NR				v			0, 4
2019 [111]		IKET	53	23.1	24	30	54												
Nicklas	USA	MET + ND	68	33.9	47	17	64	≥1	NR	3	72	64				v		v	0, 24,
2004 [112]		MET	69	34.6	59	8	67					60							72
		RC	68.5	34.4	97	44	141					72.5							
Oldham	UK	IMET (NMES)	69 ^c^	NR	17 ^c^	13 ^c^	22	1–3	NR	7	18	90 ^c^	v						0, 6,
1995 [113]		RC					8												12, 18
Raeissadat	Iran	IMET (Active) + BioF	60.2	27.6	19	2	21	1, 2	42.0	1~2	8	91.3		v					0, 8
2018 [114]		IMET (Active)	61.9	28.5	16	4	20		32.4			86.9							
Samut	USA	AET	57.6	33.9	14	0	14	2, 3	60.0 ^c^	3	6	NR				v	v	v	0, 6
2015 [115]		IKET	62.5	30.5	15	0	15												
		RC	60.9	30.4	13	0	13												
Segal	USA	LI-RET + BFR	56.1	28.7	19	0	19	1–3	≥1	3	4	97.2	v						0, 4
2015 [116]		LI-RET	54.6	32.5	21	0	21					100							
Simao	Turkey	IMET (Active)	69	27.4	9	1	10	≥2	NR	3	12	98.6					v		0, 12
2012 [117]		IMET (Active) + WBV	75	29.8	8	2	10					99.7							
		RC	71	26.7	10	1	11												
Sterzi	Italy	IMET (Active) + ND	71.3	34.8	14	9	23	≥2	81.6	3	8	90						v	0, 12
2016 [118]		IMET (Active)	71	34.3	19	8	27		86.4			90							
Tok	Turkey	IMET (NMES) + PAM	61.8	NR	16	4	20	2, 3	NR	5	3	100			v				0, 3
2011 [119]		IMET (Active) + PAM	66.6		14	6	20					100							
Varzaityte	Lithuania	AQET	63.1	29.3	52	10	62	1–3	NR	3	4	NR			v				0, 4,
2020 [120]		MET	67.9	29.8	28	2	30												8
Vassao	Brazil	MI-RET + PAM	61.6	30.5	13	0	13	2, 3	≥6	2	8	NR				v	v		0, 8
2021 [121]		MI-RET	62.3	30.1	13	0	13												
		RC	66.5	27.2	10	0	10												
Walls	Ireland	IMET (NMES)	64.4	30.7	6	3	9	3, 4	NR	3	8	99	v						0, 8
2010 [122]		IMET (Active)	63.2	32.8	4	1	5					99.4							
Walrabenstein	Netherlands	MET + ND	63.3	33.2	28	4	32	≥1	NR	2	16	NR						v	0, 8,
2023 [123]		RC	63.4	33.4	26	6	32												16
Wang	China	MET + TCM	61.4	NR	18	12	30	1, 2	39.2 ^c^	6	4	NR			v				0, 4
2016 [124]		RC	61.1		18	12	30												
Wang	China	MET + TCM	64.1	NR	24	21	45	1, 2	33.3	6	4	NR			v				0, 4
2017 [125]		RC	63.6		24	21	45		32.7										
Wang	China	IMET (Active)	57.8	NR	35	40	75	2, 3	7.8	3	6	NR				v	v		0, 6
2021 [126]		MET	58.1		39	36	75		7.5										
Wyatt	USA	AQET	40–70 ^c^	NR	NR		23	2, 3	NR	3	6	NR			v				0, 6
2001 [127]		MET					23												
Yang	China	MBT	64.5	NR	33	9	42	1–3	87	7	8	NR				v		v	0, 8
2023 [128]		PropT	64.5		34	8	42		86.2										
Yin	China	IKET + PAM	54.3	NR	23	12	35	1, 2	27.8	3	8	NR				v	v		0, 8
2021 [129]		RC	54.8		21	14	35		27.4										

^a^ Values are presented as the mean or range. ^b^ The assessed outcome measures are denoted as the letter “v”. ^c^ Data are presented as the overall mean value or pooled number of the sample. ^d^ Mean of Outerbridge osteoarthritis grade. AET, aerobic exercise training; AQET, aquatic exercise therapy; BFR, blood flow restriction; BioF, biofeedback; BMI, body mass index; CRP, C-reactive protein; CSA, cross-sectional area; HI-RET, high-intensity resistance exercise training; IKET, isokinetic exercise training; IMET, isometric exercise training; KL grade, Kellgren–Lawrence grade; KSA, the Kingdom of Saudi Arabia; LI-RET, low-intensity resistance exercise training; MBT, mind–body therapy; MET, multicomponent exercise training; MI-RET, medium-intensity resistance exercise training; MT, muscle thickness; ND, nutrition and diet; NMES, neuromuscular electrical stimulation; NR, not reported; PAM, physical agent modality; PLA, placebo intervention; PropT, proprioceptive training; RC, regular care; TCM, traditional Chinese medicine; TNF, tumor necrosis factor; TVol, thigh volume; WB-EMS, whole-body electromyostimulation; WBV, whole-body vibration.

**Table 3 biomedicines-12-01524-t003:** Identified treatment arms of included studies.

Treatment Arm		Abbreviation
Primary Exercise Training	Adjunct Treatment	
Monotherapy		
Aerobic exercise training		AET
Aquatic exercise therapy		AQET
High-intensity resistance exercise training		HIRET
Moderate-intensity resistance exercise training		MIRET
Low-intensity resistance exercise training		LIRET
Isokinetic exercise training		IKET
Isometric exercise training, self-activated muscle contraction		IMET (active)
Isometric exercise training, activated by neuromuscular electrical stimulation		IMET (NMES)
Multicomponent exercise training		MET
Mind–body therapy		MBT
Proprioceptive training		PropT
Whole-body electromyostimulation		WB-EMS
Combined treatment		
Aerobic exercise training	Blood-flow restriction	AET + BFR
Medium-intensity resistance exercise training	Physical agent modality	MIRET + PAM
Low-intensity resistance exercise training	Blood-flow restriction	LIRET + BFR
	Physical agent modality	LIRET + PAM
Isokinetic exercise training	Physical agent modality	IKET + PAM
	Traditional Chinese medicine	IKET + TCM
Isometric exercise training,	Biofeedback	IMET (Active) + BioF
self-activated muscle contraction	Nutrition and diet interventions	IMET (Active) + ND
	Physical agent modality	IMET (Active) + PAM
	Traditional Chinese medicine	IMET (Active) + TCM
	Whole-body vibration	IMET (Active) + WBV
Isometric exercise training, activated by neuromuscular electrical stimulation	Physical agent modality	IMET (NMES) + PAM
Multicomponent exercise training	Nutrition and diet interventions	MET + ND
	Physical agent modality	MET + PAM
	Traditional Chinese medicine	MET + TCM
Regular care		RC

**Table 4 biomedicines-12-01524-t004:** Associations of moderators with treatment efficiency.

Moderator	Effects on Muscle Hypertrophy ^a^						Effects on Inflammation Reduction ^a^					
	N	B	SE	Median	95% CrI		N	B	SE	Median	95% CrI	
Participant factor												
Age	26	−0.733	0.0026	−0.737	−1.446,	−0.029	30	−0.374	0.0029	−0.374	−1.169,	0.408
BMI	21	0.134	0.0028	0.124	−0.614,	0.957	21	−0.239	0.0021	−0.242	−0.813,	0.349
Sex distribution ^b^	24	−0.010	0.0028	−0.101	−0.882,	0.697	30	0.036	0.0021	0.038	−0.559,	0.621
Area of population ^c^	26	0.102	0.0025	0.109	−0.621,	0.778	30	−0.381	0.0021	−0.383	−0.989,	0.215
Disease duration	11	−0.094	0.0082	−0.119	−2.353,	2.354	13	0.165	0.0078	0.185	−1.997,	2.312
KL III-IV proportion ^d^	19	−0.539	0.0037	−0.547	−1.558,	0.509	14	−0.935	0.0175	−0.521	−7.256,	2.999
Study design factor												
ROB ^e^	26	0.296	0.0031	0.312	−0.599	1.104	30	−0.073	0.002	−0.068	−0.647,	0.481
Follow-up duration	26	−0.466	0.0015	−0.463	−0.890,	−0.040	30	0.350	0.0018	0.348	−0.144,	0.889
Intervention factor												
Treatment composition ^f^	26	0.054	0.0047	0.033	−1.293,	1.326	30	0.368	0.0018	0.372	−0.153,	0.879
Treatment duration	26	−0.451	0.0014	−0.453	−0.846,	−0.056	30	0.342	0.0019	0.334	−0.149,	0.918

^a^ Data represent the change in effects associated with the moderator indicated. B, beta coefficient; SE, standard error; 95% CrI, 95% credibility interval. ^b^ The percentage number of female participants in sample. ^c^ Code for regression model: America = 1; Asian = 2; Europe = 3. ^d^ The percentage number of participants who had Kellgren–Lawrence grade ≥ III in sample. ^e^ Code for regression model: High = 1; Unclear = 2; Low = 3. ^f^ Code for regression model: monotherapy = 1; combined treatment = 2. 95% CrI, credible interval; BMI, body mass index; ROB, risk of bias.

**Table 5 biomedicines-12-01524-t005:** GRADE certainty rating for each treatment.

Treatment (Common Comparator: RC)	Muscle Volume				Serum Level of Inflammation			
	Number of Participants (Studies)	Treatment Effect, SMD (95%CI) ^a^		Certainty of Evidence (GRADE) ^b^	Number of Participants (Studies)	Treatment Effect, SMD (95%CI) ^a^		Certainty of Evidence (GRADE) ^b^
		Direct Estimate	Network Estimate			Direct Estimate	Network Estimate	
Monotherapy								
HIRET	99 (3)	0.23 (−0.53; 1.00)	0.63 (0.01; 1.24)	⨁⨁⨁⊝ ^c^	100 (2)	−0.10 (−0.76; 0.57)	−0.15 (−0.77; 0.47)	⨁⨁⨁⊝ ^e^
MIRET	109 (3)	0.62 (0.01; 1.23)	0.70 (0.13; 1.27)	⨁⨁⊝⊝ ^cd^	131 (3)	−0.29 (−0.92; 0.35)	−0.26 (−0.88; 0.36)	⨁⨁⊝⊝ ^de^
LIRET	37 (2)		0.19 (−0.75; 1.13)	⨁⨁⨁⊝ ^e^				
AET	30 (2)	0.28 (−0.53; 1.09)	0.26 (−0.51; 1.02)	⨁⨁⊝⊝ ^ce^	139 (3)	−0.92 (−1.45; −0.38)	−0.84 (−1.35; −0.32)	⨁⨁⊝⊝ ^cd^
MET	170 (3)	−0.31 (−1.22; 0.60)	−0.31 (−1.22; 0.60)	⨁⨁⊝⊝ ^ce^	571 (8)	0.13 (−0.32; 0.57)	−0.15 (−0.50; 0.21)	⨁⊝⊝⊝ ^cde^
IMET (Active)	57 (3)	1.15 (0.02; 2.27)	0.57 (−0.22; 1.36)	⨁⨁⊝⊝ ^ce^	113 (3)	−0.60 (−1.70; 0.50)	0.36 (−0.33; 1.05)	⨁⨁⊝⊝ ^ce^
IMET (NMES)	101 (6)	0.67 (0.07; 1.27)	0.75 (0.21; 1.29)	⨁⨁⨁⊝ ^c^				
IKET	17 (1)	0.14 (−1.14; 1.42)	0.14 (−1.14; 1.42)	⨁⊝⊝⊝ ^cef^	162 (3)	−0.77 (−1.82; 0.28)	−0.40 (−1.15; 0.34)	⨁⨁⨁⊝ ^e^
AQET	85 (2)		−0.06 (−1.22; 1.10)	⨁⨁⊝⊝ ^ef^	29 (2)	−0.39 (−1.13; 0.36)	−0.39 (−1.13; 0.36)	⨁⊝⊝⊝ ^cef^
MBT	11 (1)	0.10 (−1.10; 1.29)	0.10 (−1.10; 1.29)	⨁⊝⊝⊝ ^cef^	135 (4)	−0.52 (−1.25; 0.21)	−0.60 (−1.25; 0.06)	⨁⊝⊝⊝ ^cef^
PropT					69 (2)		0.23 (−0.68; 1.14)	⨁⨁⊝⊝ ^ce^
WB-EMS					36 (1)		−0.16 (−1.09; 0.77)	⨁⨁⊝⊝ ^ef^
Combined therapy								
MIRET + PAM					13 (1)	−1.04 (−2.16; 0.08)	−0.76 (−1.76; 0.24)	⨁⨁⊝⊝ ^ef^
LIRET + PAM	19 (1)	0.00 (−1.01; 1.02)	0.12 (−0.85; 1.09)	⨁⨁⊝⊝ ^ef^				
LIRET + BFR	64 (4)	1.72 (0.87; 2.58)	1.28 (0.60; 1.97)	⨁⨁⨁⨁	16 (1)		−0.55 (−1.72; 0.61)	⨁⨁⊝⊝ ^ef^
AET + BFR	10 (1)	0.50 (−0.74; 1.75)	0.51 (−0.63; 1.66)	⨁⨁⊝⊝ ^ef^				
MET + ND	119 (1)	0.09 (−0.82; 1.00)	0.09 (−0.82; 1.00)	⨁⨁⊝⊝ ^ce^	529 (8)	−0.12 (−0.45; 0.21)	−0.19 (−0.50; 0.12)	⨁⨁⊝⊝ ^ce^
MET + PAM	30 (1)	.	0.18 (−0.97; 1.33)	⨁⨁⨁⊝ ^e^				
MET + TCM	75 (2)	0.57 (−0.14; 1.27)	0.57 (−0.14; 1.27)	⨁⨁⨁⊝^e^	92 (2)	−1.19 (−1.81; −0.58)	−1.19 (−1.81; −0.58)	⨁⨁⨁⨁
IMET (Active) + BioF	41 (2)	0.70 (−0.47; 1.88)	0.85 (−0.05; 1.75)	⨁⨁⨁⊝ ^e^				
IMET (Active) + PAM	39 (2)	0.17 (−0.83; 1.17)	0.44 (−0.31; 1.18)	⨁⨁⊝⊝ ^ce^				
IMET (Active) + ND					23 (1)		0.81 (−0.34; 1.96)	⨁⨁⊝⊝ ^ef^
IMET (Active) + TCM					106 (2)	−1.61 (−2.45; −0.78)	−1.23 (−1.85; −0.62)	⨁⨁⨁⨁
IMET (Active) + WBV					12 (1)	−1.14 (−2.25; −0.03)	−0.65 (−1.67; 0.37)	⨁⊝⊝⊝ ^cef^
IMET (NMES) + PAM	57 (4)	1.15 (0.33; 1.97)	0.88 (0.22; 1.54)	⨁⨁⨁⊝ ^c^				
IKET + PAM	20 (1)	0.09 (−0.92; 1.10)	0.21 (−0.76; 1.17)	⨁⨁⊝⊝ ^ef^	90 (2)	−1.35 (−2.25; −0.46)	−1.89 (−2.62; −1.16)	⨁⨁⨁⨁
IKET + TCM					92 (1)		−1.55 (−2.63; −0.46)	⨁⨁⨁⨁

^a^ Effect estimation is presented in standardized mean difference (SMD) with a 95% confidence interval (CI). Significant results are marked in bold. ^b^ Certainty of evidence is graded as follows: high: ⨁⨁⨁⨁; moderate: ⨁⨁⨁⊝; low: ⨁⨁⊝⊝; very low: ⨁⊝⊝⊝. ^c^ There is a high risk of bias within the studies. ^d^ The statistical heterogeneity is high (i.e., *I*^2^ > 50%). ^e^ The 95% confidence interval is wide and imprecise. ^f^ Insufficient sample size (<30). GRADE, Grading of Recommendations, Assessment, Development, and Evaluations; AET, aerobic exercise training; AQET, aquatic exercise therapy; BFR, blood flow restriction; BioF, biofeedback; HIRET, high-intensity resistance exercise training; IKET, isokinetic exercise training; IMET, isometric exercise training; LIRET, low-intensity resistance exercise training; MBT, mind–body therapy; MET, multicomponent exercise training; MIRET, medium-intensity resistance exercise training; ND, nutrition and diet; NMES, neuromuscular electrical stimulation; PAM, physical agent modality; PropT, proprioceptive training; RC, regular care; TCM, traditional Chinese medicine; WB-EMS, whole-body electromyostimulation; WBV, whole-body vibration.

## Data Availability

Refer to Appendix A. Raw data are available on request.

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
