# Peer review of "Comparative Efficacy of Various Exercise Therapies and Combined Treatments on Inflammatory Biomarkers and Morphological Measures of Skeletal Muscle among Older Adults with Knee Osteoarthritis: A Network Meta-Analysis"

_biomedicines, 2024, doi:10.3390/biomedicines12071524_

Round 1
Reviewer 1 Report
Comments and Suggestions for Authors
The authors of the manuscript entitled “Comparative Efficacy of Various Exercise Therapies and its 2 Combined Treatments on Inflammatory Biomarkers and 3 Morphological Measures of Skeletal Muscle Among Older 4 Adults with Knee Osteoarthritis: A Network Meta-Analysis” aimed to identify comparative effects among exercise monotherapy and its adjunct treatments of on muscle volume and serum inflammation for older individuals with knee osteoarthritis (KOA).
For the meta-analyses, the authors used 52 randomized controlled trials (RCTs) that combined regimens of exercise training and adjunct treatments generally achieved superior treatment effects compared with its monotherapies. Additionally, they revealed that combined treatment low-intensity resistance exercise training with blood flow restriction is most likely the optimal treatment option for muscle hypertrophy, as well as isokinetic exercise training + physical agent modality for inflammation reduction, in KOA
The study is relevant and important to guide exercise professionals working with KOA. The study was well done and the criteria for study selection were adequate, including trials with an experimental group (i.e., exercise) and a control group.
The difficulty in understanding the effects of exercise and in establishing the best exercise to prescribe is always the existence of several types of exercise, isolated or combined. In this study, the authors identified 27 different exercise treatment arms. Even when you find the same type of exercise, the intensity is often different. Besides that, the authors pointed out to best combinations of exercise for hypertrophy and inflammation and showed patient’s age may affect relative treatment efficacy.
Minor:
Figures and tables are ok, but there are too many abbreviations to follow the idea during the results and discussion section, i.e. topic 3.5 and following (page 7). Several times this reviewer had to come back to the Table 3 to check which type of exercise the authors were talking about.
Sometimes is a question of journal space but, when possible, the authors could avoid abbreviation and write the type of exercise. (i.e 3.5 first paragraph: The direct comparison results revealed that Moderate-intensity resistance exercise training, Isometric exercise training (active or activated by neuromuscular electrical stimulation) and the combination of Medium-intensity resistance exercise training + Physical agent modality or Low-intensity resistance exercise training + Blood-flow restriction obtained greater changes in muscle volume. Furthermore, the combination of low-intensity resistance exercise training with blood flow restriction exhibited superior effects on muscle morphological increases compared to exercise without blood flow restriction.
Author Response
Minor:
Figures and tables are ok, but there are too many abbreviations to follow the idea during the results and discussion section, i.e. topic 3.5 and following (page 7). Several times this reviewer had to come back to the Table 3 to check which type of exercise the authors were talking about.
Sometimes is a question of journal space but, when possible, the authors could avoid abbreviation and write the type of exercise. (i.e 3.5 first paragraph: The direct comparison results revealed that Moderate-intensity resistance exercise training, Isometric exercise training (active or activated by neuromuscular electrical stimulation) and the combination of Medium-intensity resistance exercise training + Physical agent modality or Low-intensity resistance exercise training + Blood-flow restriction obtained greater changes in muscle volume. Furthermore, the combination of low-intensity resistance exercise training with blood flow restriction exhibited superior effects on muscle morphological increases compared to exercise without blood flow restriction.
Response
Thank you for your comprehensive review and constructive comments. We understand the complexity among numerous exercise types. However, considering the consistency throughout manuscript, we decided to use abbreviations for identified exercise types. In addition, we believe that readers can easily find the full names of its corresponded abbreviations in all figures.

Reviewer 2 Report
Comments and Suggestions for Authors
This is a well-written article. Some minor suggestions are provided below:
1. Page 4, Table 2, Monotherapy column: Please provide a detailed description of items (1) resistance-based exercise training to (8) mind-body therapy. This will enable readers without a rehabilitation background to understand the content of these items. Additionally, please explain the items not mentioned in Table 2 in lines 278-284, for readers unfamiliar with rehabilitation.
2. Page 4, Table 2, Combined treatment column: What about ESWT, HA/PRP injection?
3. Line 162: Add “Outerbridge osteoarthritis grade”.
4. Lines 164-169: How did the authors determine all these cutoff points? Are there any references?
5. Line 183: Add a reference for the PEDro scale.
https://www.mdpi.com/2227-9059/11/9/2367
6. Line 190: "conflict of interest)" should be "conflict of interest"; delete the unnecessary ")".
7. Line 234: "bycalculating" should be "by calculating".
8. Line 242: Add a reference for GRADE.
https://www.mdpi.com/2075-1729/14/3/289
9. Table 2: Consider making the "country (area)" column more succinct. It is likely that none of our readers are unaware that the USA is in America and China is in Asia.
10. Line 489: Please define "impression" academically.
11. Line 499: "withdraw" should be "withdrawal".
12. Line 553: It should be "analytic results".

Minor editing of English language required
Author Response
Comments and Suggestions for Authors
This is a well-written article. Some minor suggestions are provided below:
- Page 4, Table 2, Monotherapy column: Please provide a detailed description of items (1) resistance-based exercise training to (8) mind-body therapy. This will enable readers without a rehabilitation background to understand the content of these items. Additionally, please explain the items not mentioned in Table 2 in lines 278-284, for readers unfamiliar with rehabilitation.
Response
Thank you for your comprehensive review and constructive comments. Following the reviewer’s comments, we provide a detailed description for each exercise intervention type in Table 1. In addition, in section 3.3 (lines 290-292), we explain the items not mentioned in Table 1.
- Page 4, Table 2, Combined treatment column: What about ESWT, HA/PRP injection?
Response
We removed the treatment items.
- Line 162: Add “Outerbridge osteoarthritis grade”.
Response
We revised the statement as follows:
Page 5, Line 162
“(2) characteristics of the participants including age, body mass index, sex distribution, and disease severity using Kellgren-Lawrence osteoarthritis grading system (KL grade) as well as Outerbridge osteoarthritis grade;”
- Lines 164-169: How did the authors determine all these cutoff points? Are there any references?
Response
The cutoff points for resistance training intensity were selected based on the American College of Sports Medicine's (ACSM) recommendations. We revised the statement as follows:
Page 5, Lines 169-171
“The cutoff points were selected based on the American College of Sports Medicine's (ACSM) recommendations for exercise intention in older individuals [64].”
- Line 183: Add a reference for the PEDro scale. https://www.mdpi.com/2227-9059/11/9/2367
Response
We add a reference (no. 68: Hong-Baik, I., E. Úbeda-D'Ocasar, E. Cimadevilla-Fernández-Pola, V. Jiménez-Díaz-Benito, and J.P. Hervás-Pérez, The Effects of Non-Pharmacological Interventions in Fibromyalgia: A Systematic Review and Metanalysis of Predominants Outcomes. Biomedicines, 2023. 11.) for the PEDro scale. The statement was revised as follows:
Page 5, Line 180
“The Cochrane risk of bias tool [65, 66], incorporating with the PEDro rating scale which assesses methodological quality of the analyzed studies [67], was employed to evaluate the bias risk and methodological quality of analyzed studies [68].”
- Line 190: "conflict of interest)" should be "conflict of interest"; delete the unnecessary ")".
- Line 234: "bycalculating" should be "by calculating".
Response
The statements were corrected.
- Line 242: Add a reference for GRADE. https://www.mdpi.com/2075-1729/14/3/289
Response
We add a reference (no. 80: Esteban-Sopeña, J., H. Beltran-Alacreu, M. Terradas-Monllor, J. Avendaño-Coy, and N. García-Magro, Effectiveness of Virtual Reality on Postoperative Pain, Disability and Range of Movement after Knee Replacement: A Systematic Review and Meta-Analysis. Life (Basel), 2024. 14.) for the GRADE. The statement was revised as follows:
Page 7, Line 247
“The GRADE framework has been employed to evaluate the strength of evidence in systemic reviews [80]. ”
- Table 2: Consider making the "country (area)" column more succinct. It is likely that none of our readers are unaware that the USA is in America and China is in Asia.
Response
We removed the area item in Table 2.
- Line 489: Please define "impression" academically.
Response
We revised the statement to clarify the “imprecision” as follows:
Page 21, Line 497
“The most common judgements downgrading the certainty were associated with major concerns about study limitation, imprecision (i.e. a wide range of 95% CI), …”
- Line 499: "withdraw" should be "withdrawal".
- Line 553: It should be "analytic results"
Response
The statements were corrected.

Reviewer 3 Report
Comments and Suggestions for Authors
The paper concerns an important issue: the influence of exercises on changes in serum inflammation level and muscle volume for older individuals with KOA (as a prevention of sarcopenia, too).
However, I have some comments:
1. Abstract – well structured, but not clearly written (may be shorter sentences);
2. Introduction: clear, concise;
Line 56: error in the word diabetes
3. Material/ methods: well described.
4. Results –clearly written,
5. Discussion: authors should write about limitations of the study more clearly at the end of the discussion.
6. Conclusions: too long – 3 short sentences would be enough.
7. References – quite a lot relatively new papers !
Author Response
Comments and Suggestions for Authors
The paper concerns an important issue: the influence of exercises on changes in serum inflammation level and muscle volume for older individuals with KOA (as a prevention of sarcopenia, too).
However, I have some comments:
Response
Thank you for your comprehensive review and constructive comments. Following the reviewer’s comments
- Abstract – well structured, but not clearly written (may be shorter sentences);
- Introduction: clear, concise;
Line 56: error in the word diabetes
Response
We correct the word “diabetes”.
- Material/ methods: well described.
- Results –clearly written,
- Discussion: authors should write about limitations of the study more clearly at the end of the discussion.
Response
We wrote the statements in section 4.4 (lines 588-600).
- Conclusions: too long – 3 short sentences would be enough.
Response
We rephrase the statements in Conclusions.
- References – quite a lot relatively new papers !

Round 2
Reviewer 2 Report
Comments and Suggestions for Authors
This work is suitable for publication now.
Comments on the Quality of English LanguageMinor English editing needed.